# Visualizing conformation transitions of the Lipid II flippase MurJ

Alvin C.Y. Kuk [1], Aili Hao[1], Ziqiang Guan[1] & Seok-Yong Lee [1]

The biosynthesis of many polysaccharides, including bacterial peptidoglycan and eukaryotic N-linked glycans, requires transport of lipid-linked oligosaccharide (LLO) precursors across the membrane by specialized flippases. MurJ is the flippase for the lipid-linked peptidoglycan precursor Lipid II, a key player in bacterial cell wall synthesis, and a target of recently discovered antibacterials. However, the flipping mechanism of LLOs including Lipid II remains poorly understood due to a dearth of structural information. Here we report crystal structures of MurJ captured in inward-closed, inward-open, inward-occluded and outward-facing conformations. Together with mutagenesis studies, we elucidate the conformational transitions in MurJ that mediate lipid flipping, identify the key ion for function, and provide a framework for the development of inhibitors.

[1] Department of Biochemistry, Duke University Medical Center, 303 Research Drive, Durham, NC 27710, USA. Correspondence and requests for materials should be addressed to S.-Y.L. (email: seok-yong.lee@duke.edu)

The biosynthesis of many important polysaccharides (including peptidoglycan, lipopolysaccharide, and N-linked glycans) necessitates membrane transport of oligosaccharide precursors from their cytoplasmic site of synthesis to their site of assembly outside the cytoplasm. To accomplish this task, cells utilize a translocase to attach the oligosaccharide precursor to a lipid carrier[1–3], and a flippase to transport the lipid-linked oligosaccharide (LLO) across the membrane[4]. Such LLO transport systems are evolutionarily ancient and found across all domains of life from bacteria to humans[5,6]. LLO flipping is carried out by specialized transporters of the multidrug/oligosaccharidyl-lipid/polysaccharide (MOP) superfamily, which are structurally distinct from other transporter superfamilies such as the major facilitator superfamily. MOP transporters mediate the export of numerous molecules of physiological or pharmacological importance, including LLO precursors for the biosynthesis of bacterial cell wall (MurJ, MVF family)[7–9], cell surface polysaccharides (Wzx, PST family)[10–13], eukaryotic N-linked glycosylation (Rft1, OLF family)[14–17], as well as expulsion of drug molecules (MATE family)[18].

Lipid II, undecaprenyl-diphosphate-MurNAc-pentapeptide-GlcNAc, is an essential lipid-linked peptidoglycan precursor that is large (~1900 Da), highly flexible (~70 rotatable bonds) and negatively charged (Fig. 1a). It is synthesized by the combined activities of MraY and MurG, which transfer phospho-MurNAc-pentapeptide and GlcNAc moieties to the lipid carrier undecaprenyl-phosphate (Fig. 1b)[1,19]. The cytoplasmic-facing Lipid II is flipped to the periplasmic side of the membrane by a Lipid II flippase, after which the GlcNAc-MurNAc-pentapeptide moiety is incorporated into peptidoglycan and the lipid carrier is recycled (Fig. 1b). Recent studies provide strong evidence that MurJ, a MOP superfamily member, is the long-sought-after Lipid II flippase[7–9]. Consistent with this idea, in vitro binding of Lipid II to MurJ from Escherichia coli has recently been demonstrated by native mass spectrometry[20]. Studies also show that MurJ is a key player in bacterial cell division[21] and is the target of newly-discovered inhibitors, such as the humimycins[22,23] and the phage M lysis protein[24], both of which inhibit MurJ function to exert their bactericidal effects. Beyond bacterial physiology and antibiotic development, MurJ is important as a prototype LLO flippase and provides a framework for understanding the LLO flipping mechanism.

We previously determined a crystal structure of MurJ from Thermosipho africanus (MurJ$_{TA}$), which exhibits an inward-facing conformation[25]. MurJ consists of 14 transmembrane helices (TMs), of which the first 12 are organized into two 6-TM bundles termed the N-lobe (TMs 1–6) and C-lobe (TMs 7–12) (Fig. 1c)[25]. The N-lobes and C-lobes of MOP superfamily transporters are related by pseudo-twofold rotational symmetry with axis normal to the membrane, although in this MurJ structure the symmetry is notably distorted between TM 1 and 7, as well as between TM 2 and TM 8 (Fig. 1d)[25]. The N-lobe and C-lobe enclose a central cavity that is large and highly cationic, properties consistent with Lipid II transport. TMs 13–14, not part of the core transport domain, form a hydrophobic groove that connects to the central cavity through a membrane portal. Based on this structure and previously reported chemical genetics data[7], we proposed that MurJ transports Lipid II from the inner to the outer leaflet of the membrane using an alternating-access mechanism[25] distinct from what has been proposed for other lipid transporters[26,27]. Recent cysteine-crosslinking data in E. coli cells also support our alternating-access model of Lipid II transport by MurJ[28].

Transporters which utilize an alternating access mechanism typically undergo multiple conformational states, including inward-facing open, inward-facing occluded, outward-facing occluded, and outward-facing open[29,30]. Because only one state has been previously visualized in MurJ, the design principles and conformation transitions of flippases in the MOP superfamily are poorly understood[25]. Furthermore, the questions of whether MurJ is an ion-coupled transporter and which ion(s) are involved remain unresolved. While some drug exporters in the MOP superfamily have been characterized to be Na$^+$ or H$^+$ coupled[12,31–33], our high-resolution inward-facing MurJ structure did not reveal a cation that could be important for transport[25]. It has been shown that MurJ activity is dependent on membrane potential and not on a H$^+$ gradient[34], but the involvement of Na$^+$ has not been ruled out. In particular, it was found that dissipating the membrane potential decreases prevalence of the inward-facing but not the outward-facing state of E. coli MurJ (MurJ$_{EC}$)[28].

Our structures of MurJ in multiple stages of its transport cycle, together with in vitro cysteine accessibility and mutagenesis data, shed light on the conformation transitions that mediate Lipid II flipping in bacterial peptidoglycan biosynthesis.

## Results

### Structure determination of MurJ$_{TA}$ in multiple conformations.
Because we observed a chloride ion in the central cavity of our previous crystal structure, we hypothesized that chloride might be restricting MurJ$_{TA}$ in this crystal form and took precautions to remove chloride from cell lysis and protein purification buffers. With this optimized protein purification procedure and in the presence of Lipid II doped into the lipidic cubic phase, we obtained additional crystal forms of MurJ$_{TA}$ (Supplementary Fig. 1a). We determined four crystal structures of MurJ$_{TA}$: one structure in an outward-facing conformation to 1.8 Å resolution, and three structures in inward-facing conformations distinct from the published structure, resolved to 3.2, 3.0, and 2.6 Å, respectively (Fig. 1e). We refined the structures to good geometry and minimal clashes (Supplementary Table 1). The final electron density (Supplementary Fig. 1b) allowed almost all residues to be built unambiguously except for a few residues at the termini.

### Inward-facing structures reveal a lateral-gating mechanism.
Superposition of all the inward-facing structures show that they align well at the periplasmic end of the protein (Fig. 2a), which highlights the importance of the periplasmic gate in stabilizing the inward-facing state. This gate is formed by hydrogen-bond interactions between the N-lobe and C-lobe (Fig. 2b). In contrast to the periplasmic gate, the structures diverge substantially on the cytoplasmic side (Fig. 2c, d). Coordinated rearrangement of TM 1 (blue), TM 8 (green), and the loop between TM 4 and TM 5 (TM 4–5 loop) regulates the size of the membrane portal (Fig. 2c, d). Based on the conformation at these regions and the size of the membrane portal, we assigned two of the structures as inward-closed and inward-open. The Cα-Cα distance between Ser11 (TM 1) and Ser267 (TM 8) increased from 8.0 Å in the closed structure to 17.4 Å in the open structure. TM 1 is bent outwards by almost 40° in the open structure about the invariant Gly21, coordinated by rearrangement of the TM 4–5 loop and unwinding of the cytoplasmic ends of TM 4 and TM 5 (Fig. 2d). The conserved Phe151 (Phe157 in MurJ$_{EC}$) at the TM 4–5 loop provides leverage to bend TM 1 out into the membrane. TM 8, which contains several non-α helical (π helical or 3$_{10}$ helical) elements, undergoes dynamic changes that allow hinge bending and register shift (Fig. 2c). Because Lipid II is believed to access the central cavity of MurJ through the portal between TM 1 and TM 8[25], these changes in the portal could regulate the entry of the large Lipid II headgroup into the central cavity via a lateral-gating mechanism. Residues in TM 1 (Arg18 and Gly21) as well as those

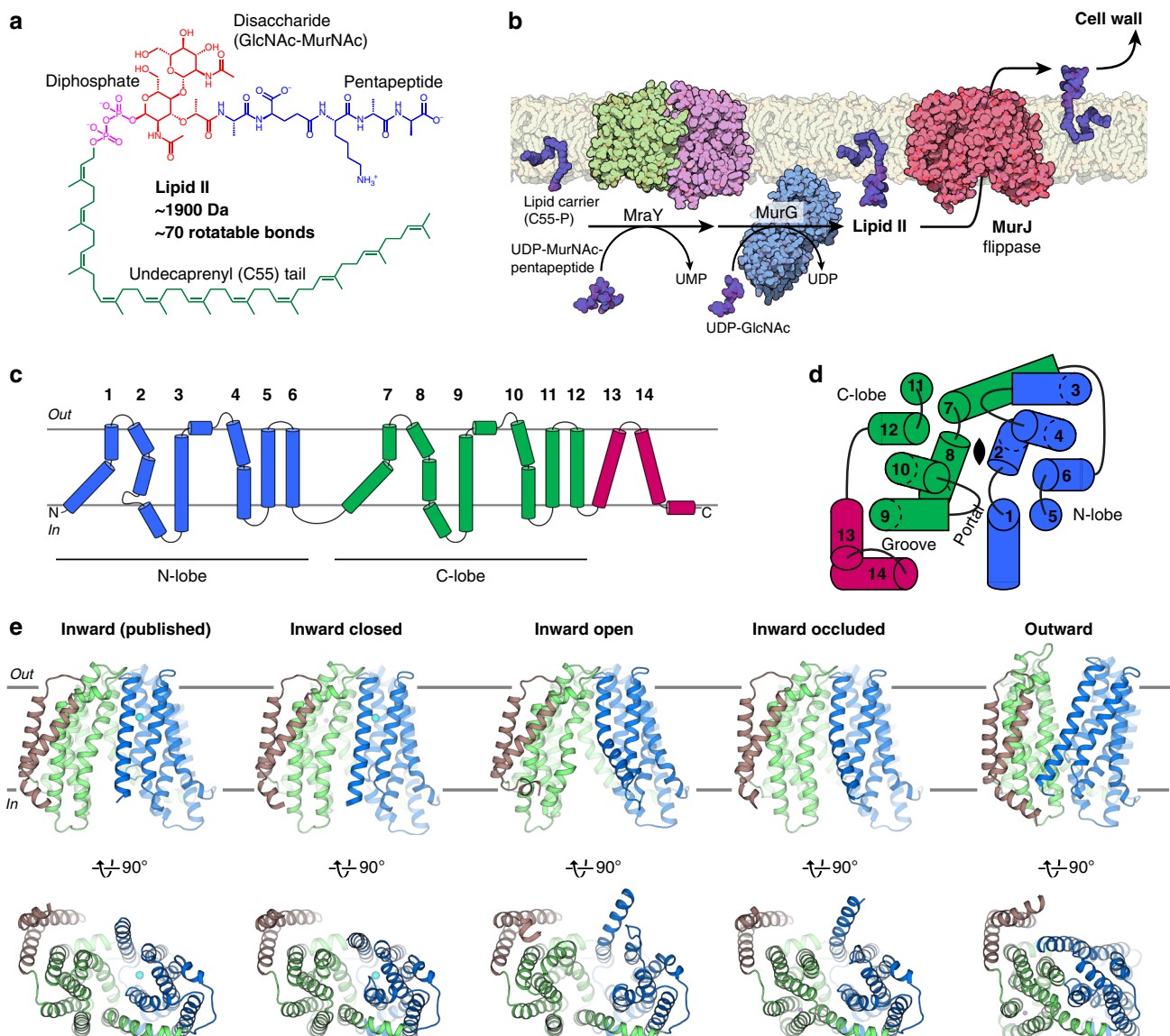

**Fig. 1** Crystal structures of the Lipid II flippase MurJ_TA. **a** Chemical structure of the peptidoglycan precursor Lipid II. The third residue of the pentapeptide is variable (shown here is L-lysine but could be meso-diaminopimelate in many Gram-negative bacteria). **b** Lipid II is synthesized by the combined transferase activities of MraY (PDB 4J72) and MurG (1F0K), then flipped across the cytoplasmic membrane by MurJ for subsequent incorporation into the cell wall. These steps constitute a prototype lipid-linked oligosaccharide (LLO) transport system. **c** MurJ contains two 6-helix bundles (N-lobe, blue, and C-lobe, green) common to the MOP superfamily and 2 additional C-terminal helices (brown). **d** N-lobe and C-lobe are related by pseudo-twofold rotational symmetry normal to the membrane (top view is shown). Symmetry is distorted at TMs 1, 2, 7, and 8 which enclose the central cavity. TMs 13 and 14 form a hydrophobic groove that enters the central cavity through a portal. **e** Our previous crystal structure of MurJ (5T77), and crystal structures in the inward-closed, inward-open, inward-occluded, and outward conformations (front and bottom views). Resolutions (from left) are 2.0, 3.2, 3.0, 2.6, and 1.8 Å

in the TM 4–5 loop (Leu145, Asn146, Phe151, and Pro158) are conserved in MurJ but not in multidrug and toxic compound extrusion (MATE) drug efflux pumps in the same superfamily (Supplementary Fig. 2), suggesting the putative lateral gate could be a specific adaptation for flippase function and thus a design principle for a LLO flippase.

To probe the in vitro conformational states of MurJ_TA, we designed single cysteine mutants at the cytoplasmic and periplasmic gates. Because MurJ_TA has no endogenous cysteines, these mutants would manifest an upward shift in mass if the cysteine was accessible when probed by PEG-maleimide. We observed that single cysteine mutants at the cytoplasmic gate were much more accessible to PEG-maleimide than those at the periplasmic gate, suggesting that MurJ_TA predominantly adopts inward-facing conformations in detergent micelles

(Supplementary Fig. 3). We also performed the experiments at an elevated temperature (60 °C) because MurJ_TA is from a hyperthermophile bacterium. We observed some changes in the accessibility profile at 60 °C including reduction of cysteine accessibility at the residues comprising the cytoplasmic gate on TM 8 (Thr270, Ser274, and Ser277), indicating that MurJ_TA in detergent micelles is sampling multiple conformations.

**Asymmetric inward-occluded conformation of MurJ_TA.** One of the inward-facing structures exhibited a partially dilated membrane portal but was otherwise divergent from the rest of the inward-facing structures (Fig. 3a). Specifically, the C-lobe is bent with the cytoplasmic section swung ~15° towards the N-lobe. Conserved residues Pro260, Pro300, and Gly340 serve as hinges

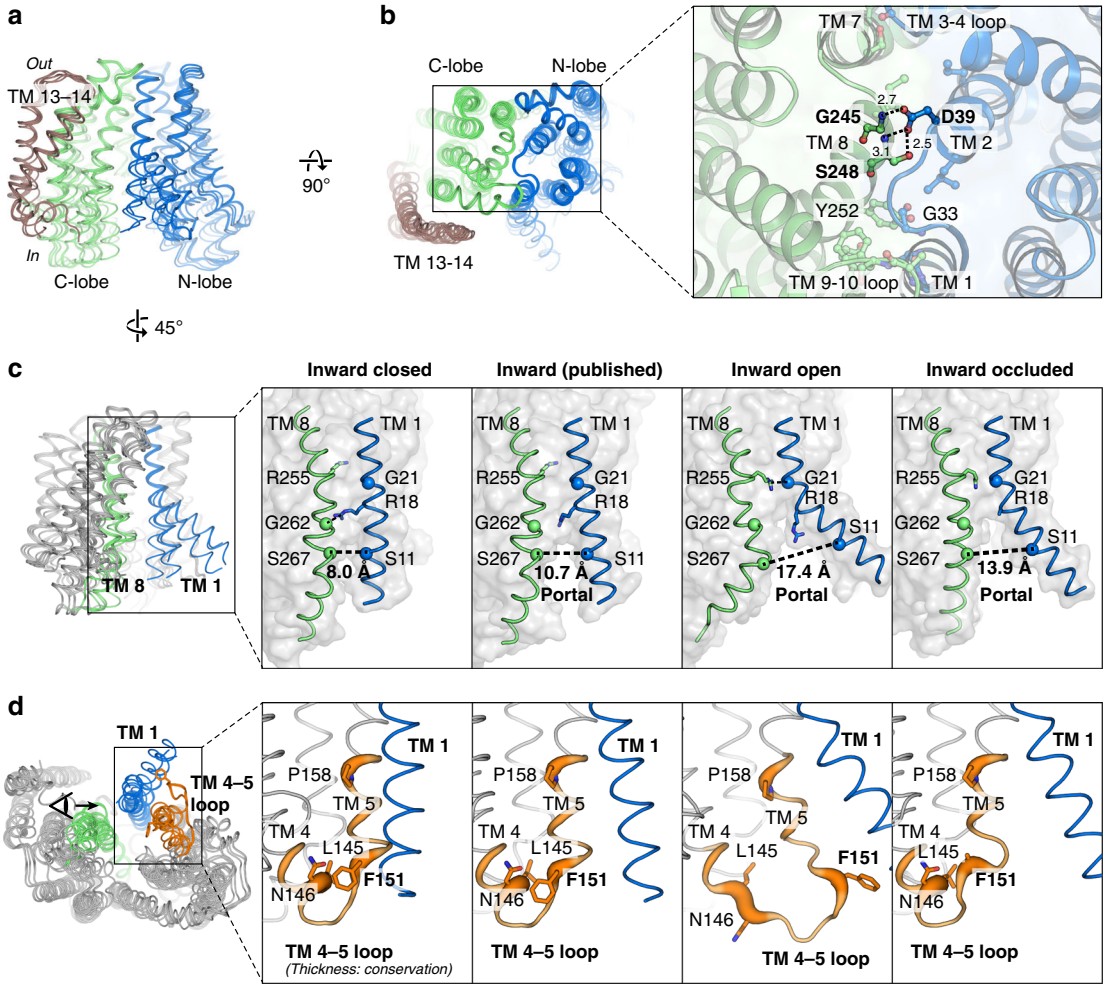

**Fig. 2** Lateral gating mechanism in the inward-facing state. **a** Superposition of inward-facing structures highlights similarity at the periplasmic gate but divergence at the cytoplasmic gate. **b** The periplasmic gate is stabilized by hydrogen bonds between Gly245/Ser248 and the essential Asp39. **c** At the cytoplasmic side, coordinated rearrangement of TM 1 (blue) and TM 8 (green) controls portal dilation. The Cα-Cα distance between Ser11 (TM 1) and Ser267 (TM 8) increases from 8.0 Å in the inward-closed structure to 17.4 Å in the inward-open structure. **d** Rearrangement of the TM 4–5 loop (orange) and unwinding of the cytoplasmic ends of TMs 4/5 could induce bending of TM 1. The conserved Phe151 (Phe157 in MurJ$_{EC}$) provides leverage to bend TM 1 (residues 1–20) out into the membrane. TM 4–5 loop is shown in sausage representation, with thickness proportional to conservation. Sidechains of Arg18 and Phe151 were not resolved in the inward-occluded structure

on TMs 8, 9, and 10, respectively. In contrast, re-arrangements in the N-lobe are restricted to the middle section of TM 2, which also bends inwards towards the central cavity. These conformational changes appear to be aided by the flexible cytoplasmic half of TM 8, which assumes a distinctly non-α-helical conformation, as well as the broken helix in TM 2 formed by the G/A-E-G-A motif that is conserved in Gram-negative MurJ sequences (Fig. 3b). Together these changes position Glu57 from the G/A-E-G-A motif into proximity (~ 4 Å) of Arg352 on TM 10 (or Lys368 one helical-turn up in MurJ$_{EC}$), forming a thin gate between the N-lobe and C-lobe (Fig. 3a, b). Notably, we observed unmodeled electron density peaks at the portal and at the central cavity above the thin gate (Fig. 3c). Although we cannot unambiguously assign the electron density to Lipid II, we hypothesized that this Glu57-Arg352 thin gate could occlude the flexible Lipid II molecule into the cavity prior to outward transition. We performed in silico docking and molecular dynamics (MD) simulation of the docked MurJ-Lipid II complex in a hydrated phospholipid bilayer system, which shows that the inward-occluded cavity could fit the Lipid II headgroup above the thin gate (Supplementary Fig. 4a–c).

To test the importance of this thin gate to MurJ function, we mutated Arg352 to either Ala or Gln, both of which resulted in loss of complementation, albeit with some reduction of expression (Supplementary Figs. 5 and 6). Together with our previous mutagenesis study on Glu57[25], this indicates that Glu57 and Arg352 are important for MurJ$_{TA}$ function and/or folding, and suggests that electrostatic interaction might be the major contributor to thin gate stability. Similarly, loss of fitness was observed for the corresponding Lys368Glu and Lys368Ile mutants in MurJ$_{EC}$ upon re-examination of previous mutagenesis data, but not Lys368Gln, suggesting that the putative thin gate could be stabilized by a hydrogen bonding interaction in MurJ$_{EC}$ (Supplementary Table 2)[35]. While conservation of mechanistic detail at the thin gate remains to be determined, our results are consistent with the idea that the thin gate is important for occluding the Lipid II headgroup from the cytoplasm (Supplementary Fig. 4d). However, we caution against excessive interpretation of our complementation data on MurJ$_{TA}$ as well as previous mutagenesis data on MurJ$_{EC}$ as these experiments cannot determine the exact functional roles of these residues. Furthermore, a reliable in vitro lipid II flipping assay for MurJ is

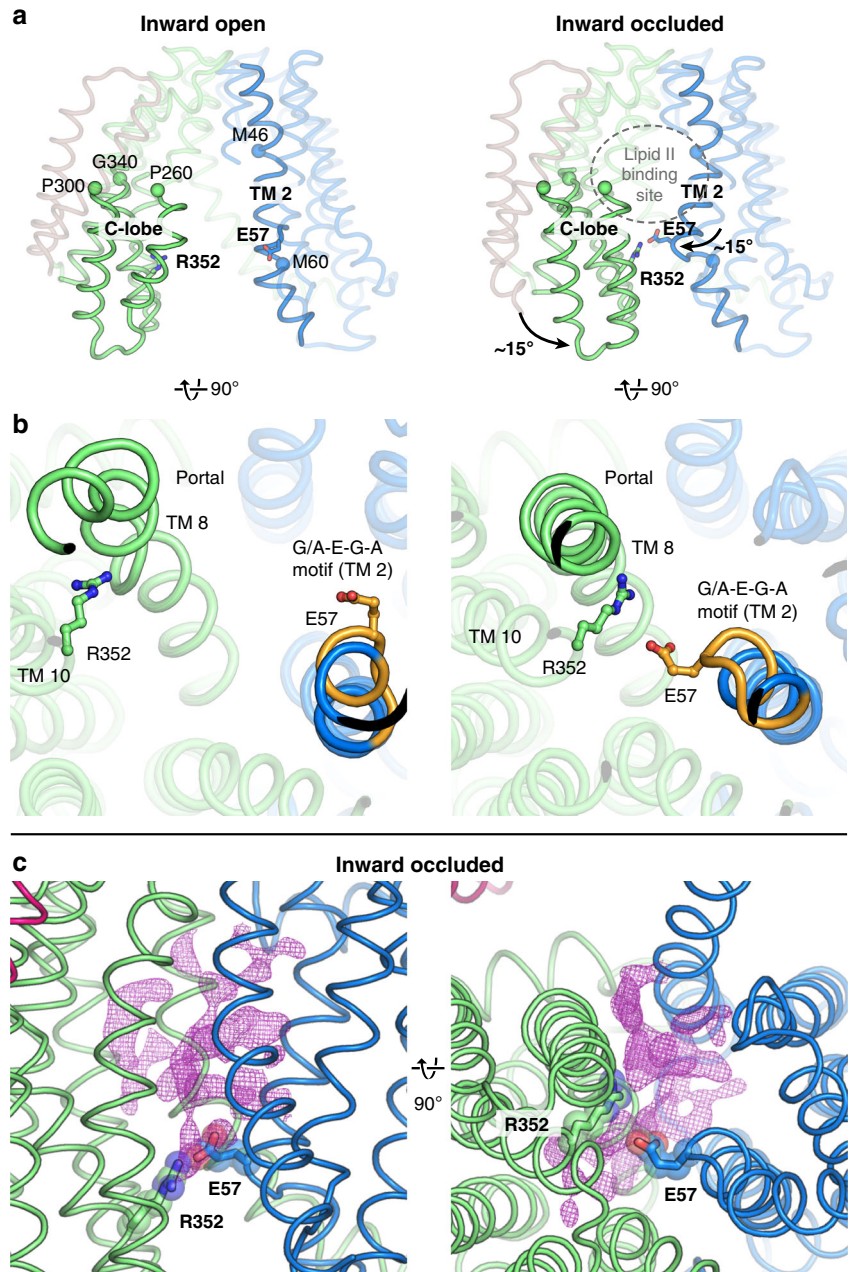

**Fig. 3** The asymmetric inward-occluded conformation of MurJ$_{TA}$. **a** Rotation of the cytoplasmic half of C-lobe by ~15° from the inward-open structure (left) to the inward-occluded structure (right). Conserved residues Pro260, Pro300, and Gly340 serve as hinges. At the same time, the middle segment of TM 2 also bends inwards by ~15°. Together these motions bring Glu57 (TM 2) and Arg352 (TM 10, Lys368 in *E. coli* MurJ) into proximity, which form a thin gate. Most of TM 1 is hidden for clarity. **b** The conserved (G/A)-E-G-A motif (orange) allows S-shaped bending of TM 2 by breaking the helix. TM 8 assumes a more α-helical geometry (albeit still not ideal) in the inward-occluded structure than in the other inward structures. **c** Unmodeled 2Fo − Fc electron density at the portal and transport cavity of the inward-occluded structure, displayed here as purple mesh contoured to 0.8 σ

still not available despite attempts by us and others. Based on the abovementioned conformational changes, presence of the putative thin gate, and the unmodeled electron density, we tentatively assign this structure as an inward-occluded state of MurJ. Inward-open-to-inward-occluded transition appears to occur asymmetrically with a larger rotation of the C-lobe for thin gate formation than the N-lobe.

**Outward-facing conformation of MurJ$_{TA}$.** Our high-resolution outward-facing structure allows for structural comparison with the inward-facing occluded structure to determine the

conformational changes that could drive outward transition. The outward-facing cavity was substantially shallower and narrower than the inward-facing cavity, and thus cannot accommodate the head group of Lipid II. This suggests that our outward facing structure represents a state in which the head group of Lipid II is released into the periplasm, consistent with the idea that cavity shrinkage could be a mechanism to displace substrate into the periplasm[33,36] (Fig. 4a). The inward-to-outward state transition is associated with a rotation of the N-lobes and C-lobes resulting in cytoplasmic gate closure and periplasmic gate opening (Fig. 4b). The N-lobe rotated by ~15° from the inward-occluded structure while the C-lobe rotated by only ~7.5°, possibly because the latter

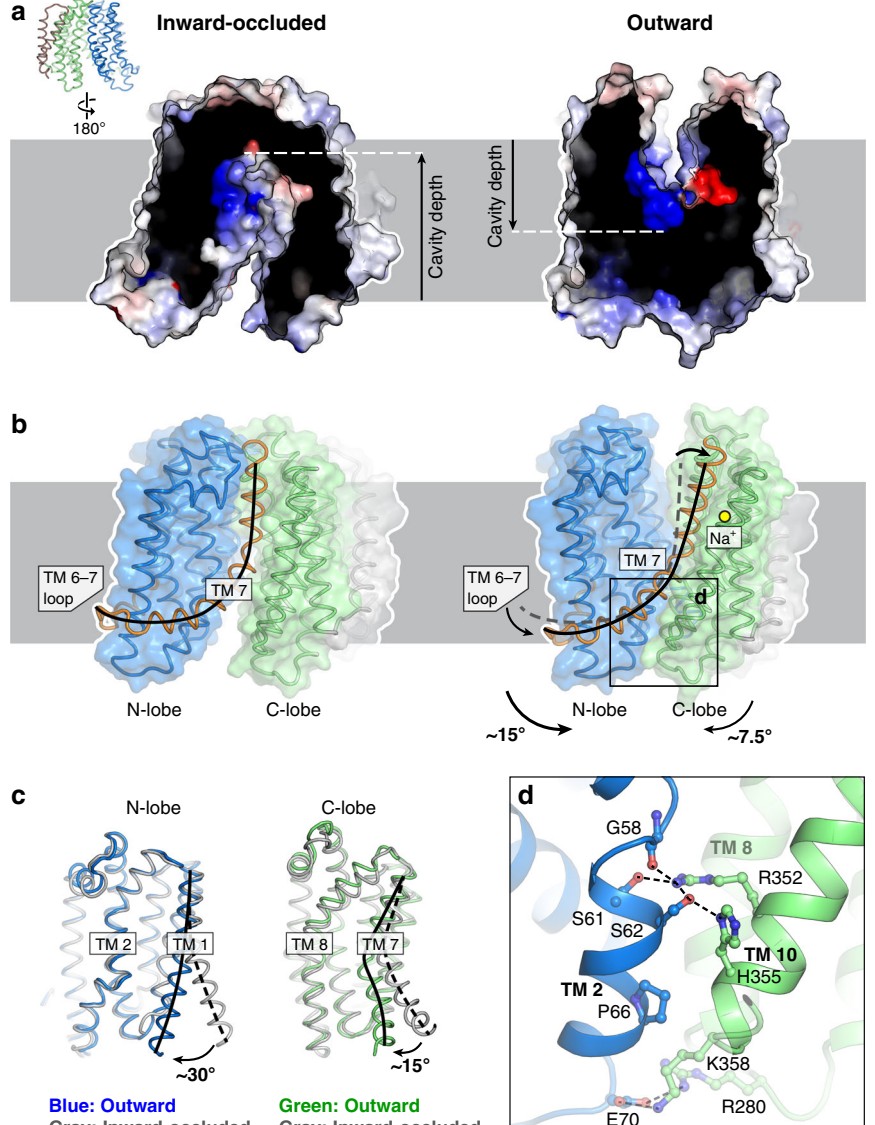

**Fig. 4** The outward conformation of MurJ$_{TA}$. **a** The outward-facing cavity is more shallow and narrow than the inward-facing cavity. **b** Straightening of TM 7 and concomitant lowering of the TM 6–7 loop could close the cytoplasmic gate, allowing transition from the inward-occluded to the outward-facing conformation. **c** Alignment of N-lobes and C-lobes from the inward-occluded conformation (gray) to those of the outward-facing conformation (in color), showing rotation of not just TM 7 but also TM 1, the latter closing the lateral membrane portal. **d** The cytoplasmic gate is mainly stabilized by a hydrogen bond network between TMs 2 and 10, with less-conserved electrostatic interactions at the cytoplasmic loops

has swung inwards in the asymmetric occluded structure (Fig. 4b). The fulcrum of this rotation was located at the TM 7 bridging the N-lobes and C-lobes, which was bent by 90° in the inward-occluded structure but straightened in the outward-facing structure. This results in concomitant lowering of the TM 6–7 loop, which is amphipathic and partially embedded in the cytoplasmic leaflet of the membrane as it loops around the N-lobe. We postulate that tension from TM 7 straightening would lower the TM 6–7 loop in the membrane, providing the force required for N-lobe rotation and closure of the cytoplasmic gate. TM 1 also straightens from its bent conformation in the inward-occluded structure, closing the lateral membrane portal (Fig. 4c). We observed unmodeled density in the hydrophobic groove leading into the outward-facing cavity, which could be the binding site of the undecaprenyl lipid tail of Lipid II while the headgroup is in the outward-facing cavity (Supplementary Fig. 7). Below the shallow outward-facing central cavity is the cytoplasmic gate which is ~20 Å thick and formed by TMs 2, 8, and 10 (Fig. 4d).

This gate is stabilized by hydrogen bonds between TM 2 and TM 10, again involving Arg352. Notably, Arg352 forms the thin gate with Glu57 in the inward-occluded structure, but now interacts with Ser61, Ser62, and the backbone carbonyl of Gly58 in the G/A-E-G-A helix break, suggesting that Arg352 plays a key role in conformational transition from the inward-facing to the outward-facing states. Our outward-facing structure is consistent with accessibility studies done on *E. coli* MurJ (MurJ$_{EC}$)[24,28,34,37], underscoring a shared transport mechanism between MurJ$_{TA}$ and MurJ$_{EC}$ (Supplementary Fig. 8).

**The sodium site in MurJ$_{TA}$.** We noticed a strong Fo − Fc omit electron density peak (~19 σ) in the C-lobe between TMs 7, 11, and 12 (Fig. 5a). The coordination distances were within the expected range for Na$^+$ (2.3–2.5 Å) in our 1.8 Å-resolution-structure and the coordination geometry is most consistent with Na$^+$, as determined by the CheckMyMetal server[38]. Also, Na$^+$ is

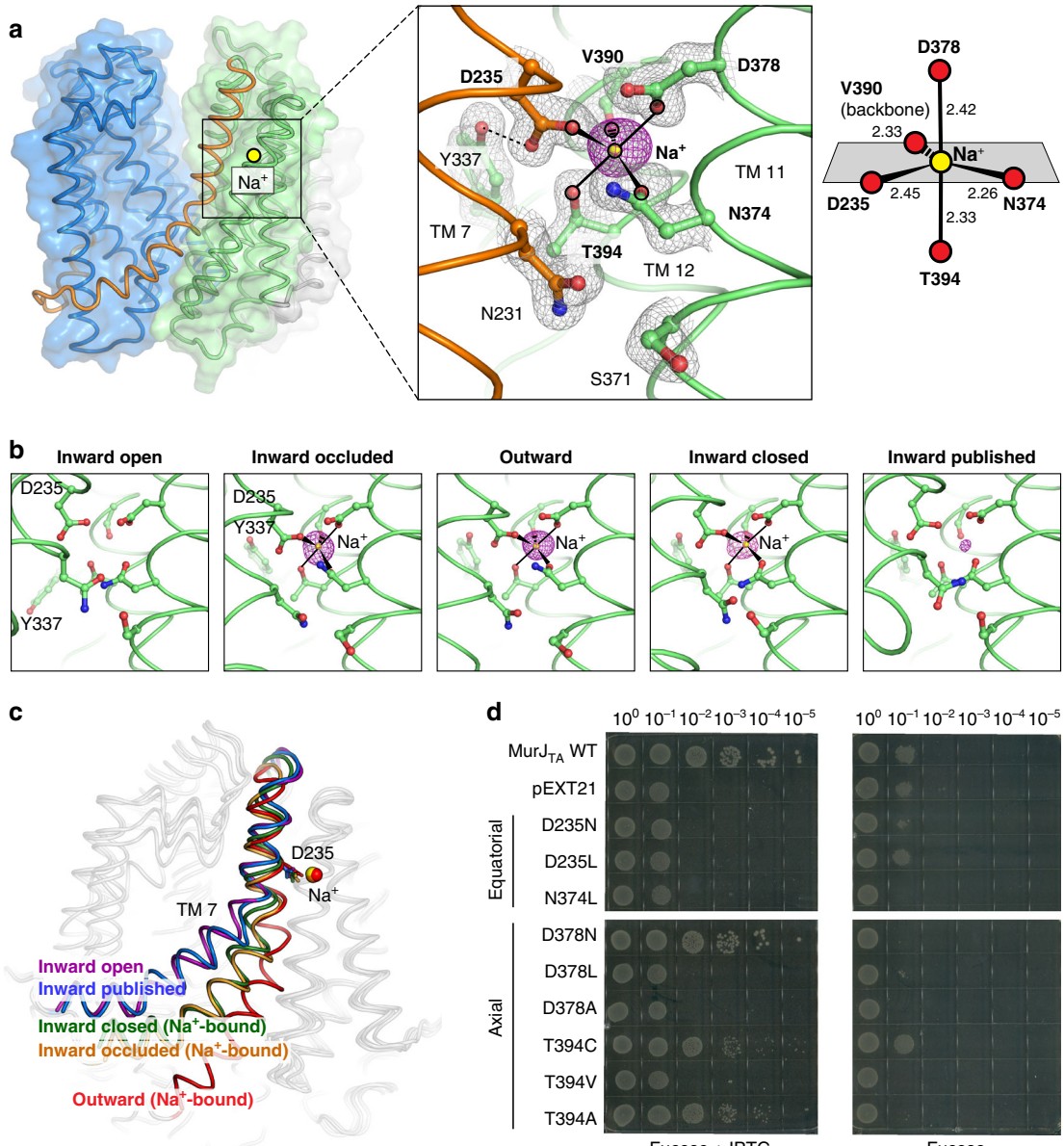

**Fig. 5** The sodium binding site in MurJ$_{TA}$. **a** Na$^+$ is bound in the C-lobe (TMs 7, 11, 12) coordinated by Asp235, Asn374, Asp378, Val390 (backbone carbonyl), and Thr394. Fo − Fc omit density for Na$^+$ is shown as magenta mesh contoured to 4.5 σ (peak height ~19 σ, 1.8 Å resolution). 2Fo − Fc omit density is shown in gray mesh. Na$^+$ is coordinated by trigonal bipyramidal geometry. **b** Na$^+$ was also bound in the inward-occluded and inward-closed structures. **c** Na$^+$-associated rearrangement could be propagated down TM 7. **d** Complementation assay of MurJ$_{TA}$ sodium site mutants in MurJ$_{EC}$-depletion strain NR1154. Cells transformed with plasmids encoding MurJ$_{TA}$ (wild-type or sodium site mutant) or without insert (pEXT21) were depleted of endogenous MurJ by serial dilution on agar plates containing the anti-inducer D-fucose. MurJ$_{TA}$ expression was induced by addition of IPTG. Data shown are representative of three biological replicates. Source data are provided as a Source Data File. Mutants D235A and N374A were previously determined to neither express nor complement[25]

the only cation present in one of the crystallization conditions that yielded the outward-facing state (see Materials and methods section). Based on these criteria, we assigned this peak as Na$^+$. Sodium is coordinated in trigonal bipyramidal geometry, with an equatorial plane formed by Asp235, Asn374, Val390 (backbone carbonyl) and the axial positions occupied by Thr394 and Asp378. Asp235 is the only Na$^+$ coordinating residue from TM 7; the others are found in either TM 11 or TM 12. Notably, the DNDXT amino acid composition (where X = backbone carbonyl) and coordination geometry of this Na$^+$ site is completely conserved with that recently identified in MATE transporters, although it is located in the N-lobe of the latter[39] (Supplementary

Fig. 9). Furthermore, the Na$^+$-coordinating oxygen atoms are conserved in MurJ$_{EC}$[35], although the Na$^+$-bound MurJ$_{EC}$ structure is not available (Supplementary Fig. 10).

In light of this discovery, we re-analyzed the Na$^+$ site in all our other MurJ structures and found electron density in the inward-occluded and inward-closed structures as well, albeit much weaker (Fig. 5b and Supplementary Table 3). Because of the high concentration of Na$^+$ in the crystal drop outweighing the potentially lower Na$^+$ affinity in the inward-facing state, we posit that these weak densities would correspond to Na$^+$, possibly reflecting the inward-facing pre-Na$^+$-release conformation of MurJ. Comparison of our three Na$^+$-bound structures revealed

different degrees of TM 7 straightening, resulting in concomitant lowering of the N-terminal half of TM 7 and the TM 6–7 loop relative to the $Na^+$ site, suggesting that $Na^+$-induced conformational change at Asp235 could be propagated down TM 7 (Fig. 5c). The outward-facing structure displays the most straightening of TM 7 and the most optimal $Na^+$ coordination geometry (Supplementary Table 3).

Guided by our structures, we performed mutagenesis on the sodium-coordinating residues and assayed their effect on $MurJ_{TA}$ function as previously described[8,25]. None of the mutants at the equatorial positions were able to complement, whereas certain substitutions at axial positions were tolerated (Fig. 5d). We note that the mutants failing to complement also failed to express based on analysis of the total membrane fraction by Western blot (Supplementary Fig. 6), suggesting that substitutions at the sodium site would perturb protein folding, which precludes mutagenesis approaches for studying the role of $Na^+$ in MurJ function. Sodium-dependent conformational changes were previously observed in the MATE transporter NorM by spectroscopic studies[32]. Our mutagenesis experiments showed that this crystallographically-identified $Na^+$ site is essential for MurJ folding. However, it remains to be determined whether the observed $Na^+$ works as a counter-transporting ion for Lipid II flipping.

**State-dependent changes in the central cavity.** The central cavity of $MurJ_{TA}$ undergoes state-dependent changes in its size and position with respect to the membrane. The cavity was largest in the inward-open state, was located closer to the periplasmic side in the inward-occluded state, and resembled a narrow crevice in the outward-facing state (Fig. 6a). We observed systematic rearrangement of Arg24, Asp25, and Arg255 (Arg270 in $MurJ_{EC}$) in the cavity between the different crystal structures (Fig. 6b). These three residues are conserved with Arg24 being invariant. Both Arg24/Arg255 are essential[25,37,40], while Asp25 is intolerant of certain substitutions[35]. This Arg24-Asp25-Arg255 triad is held in position by Asn162, Gln251, and a chloride ion in the inward-closed state. Arg255 is released from Gln251 in the inward-open state and swivels down, possibly to capture the diphosphate moiety of Lipid II that enters through the portal. In the inward-occluded state, Arg24 and Arg255 are brought into unusual proximity (just 2.9 Å apart at the closest) on top of the unmodeled electron density. Since such close contact between two arginine side chains would experience unfavorable repulsion, we reasoned that they could be stabilized by electrostatic interactions with Asp25 and the diphosphate moiety of Lipid II. The periplasmic gate opens in the outward conformation, disengaging Arg255 and allowing substrate release. Because of their conserved and essential nature, as well as their coordinated rearrangement at the apex of the inward-facing cavity in proximity to the unmodeled electron density, we suggest Arg24-Asp25-Arg255 to be the putative substrate-binding triad.

We next tested whether these residues are important for $MurJ_{TA}$ function by performing a mass spectrometry assay for cellular Lipid II accumulation (Supplementary Fig. 11). Since MurJ is essential in *E. coli*, depletion of MurJ results in defect in cell growth and eventually cell death[8]. Expression of wild-type $MurJ_{TA}$ rescued cell growth but not mutants R24A and R255A (Supplementary Fig. 11a), consistent with our previous results indicating that these two mutants are non-functional[25]. Because it has been reported that Lipid II accumulates when MurJ activity is abolished[34,41], we directly examined cellular Lipid II levels in these cells. Lipid II levels were approximately five-fold higher in cells expressing the R24A or R255A $MurJ_{TA}$ mutants as compared to that in wild-type $MurJ_{TA}$ (Supplementary Fig. 11b), which is consistent with importance of these residues in Lipid II transport.

## Discussion

We determined the conformational landscape of the lipid flippase MurJ, and our analyses suggest that MurJ undergoes an ordered trajectory through its transport cycle (Fig. 6c). We believe that the transport mechanism of $MurJ_{TA}$ is conserved in general with that of other orthologs including $MurJ_{EC}$ for the following reasons. (1) The inward-facing structures of $MurJ_{TA}$ and $MurJ_{EC}$ are very similar despite low sequence identity[25,35]; (2) Our outward-facing structure is consistent with accessibility (Supplementary Fig. 8) and mutagenesis studies (Supplementary Table 2 and Supplementary Fig. 12) studies done on $MurJ_{EC}$ by many groups[24,28,34,37]; (3) conserved cationic residues in the central cavity of $MurJ_{TA}$ are important for function consistent with previous mutagenesis studies on $MurJ_{EC}$[25,35,37,40]. However, we also acknowledged that the specific mechanistic details could vary in different MurJ orthologues, as is common in transporters[30,42] (Supplementary Table 2 and Supplementary Fig. 12). We caution against excessive mechanistic interpretation of previous experimental results done on $MurJ_{EC}$ as well as our complementation results based on the $MurJ_{TA}$ structures, because multiple factors (different in vivo experiments, different orthologs, and lack of in vitro experiments) could make mechanistic interpretations of these data non-trivial.

Our study highlights the importance of dynamic conformation changes to the mechanism of Lipid II flipping, which could be a principle for LLO flipping by LLO flippases in general. In particular, TM 8 undergoes dynamic conformational changes through transient π or $3_{10}$ helical elements throughout the transport cycle. These dynamic helical secondary structure changes were proposed to be important for gating of Transient Receptor Potential ion channels[43,44], but their presence in our $MurJ_{TA}$ structures suggest that they are more universal for membrane transport proteins than were previously recognized.

The outward-facing structure of $MurJ_{TA}$ indicates importance of the cytoplasmic gate in occluding access to the outward-facing cavity from the cytoplasm. Because we observed substantial divergence at the cytoplasmic gate among the inward-facing structures, dynamic conformation changes at this gate appear to be important for MurJ function. Consistent with this finding, resistance mutants to the humimycins, lipopeptides with antibacterial activity against some Gram-positive pathogens including methicillin-resistant *Staphylococcus aureus*[22], map mostly to the cytoplasmic side of MurJ below the putative lipid II headgroup binding site[23]. While the mechanism of inhibition by humimycins is not currently known, it is conceivable that they could disrupt closure of the cytoplasmic gate and thus trap MurJ in one of the inward-facing conformations.

These structures of $MurJ_{TA}$ also demonstrate the importance of dynamic TM 7 rearrangement for inward to outward transition. Resistant mutants to the levivirus M lysis protein ($Lys^M$), which induces bacterial cell lysis by inhibition of MurJ function, was previously mapped to the TM 2-TM 7 region that is facing outside towards the membrane[24]. Cysteines positioned in the cavity displayed increased MTSES accessibility in cells upon $Lys^M$ treatment, suggesting stabilization of the outward-facing state upon $Lys^M$ binding[24,34], consistent with the role of TM7 in controlling the conformational state of MurJ. As exemplified by humimycins and $Lys^M$, our study provides a framework for understanding how the conformational dynamics of a LLO flippase could be exploited by natural product inhibitors for antibiotic targeting.

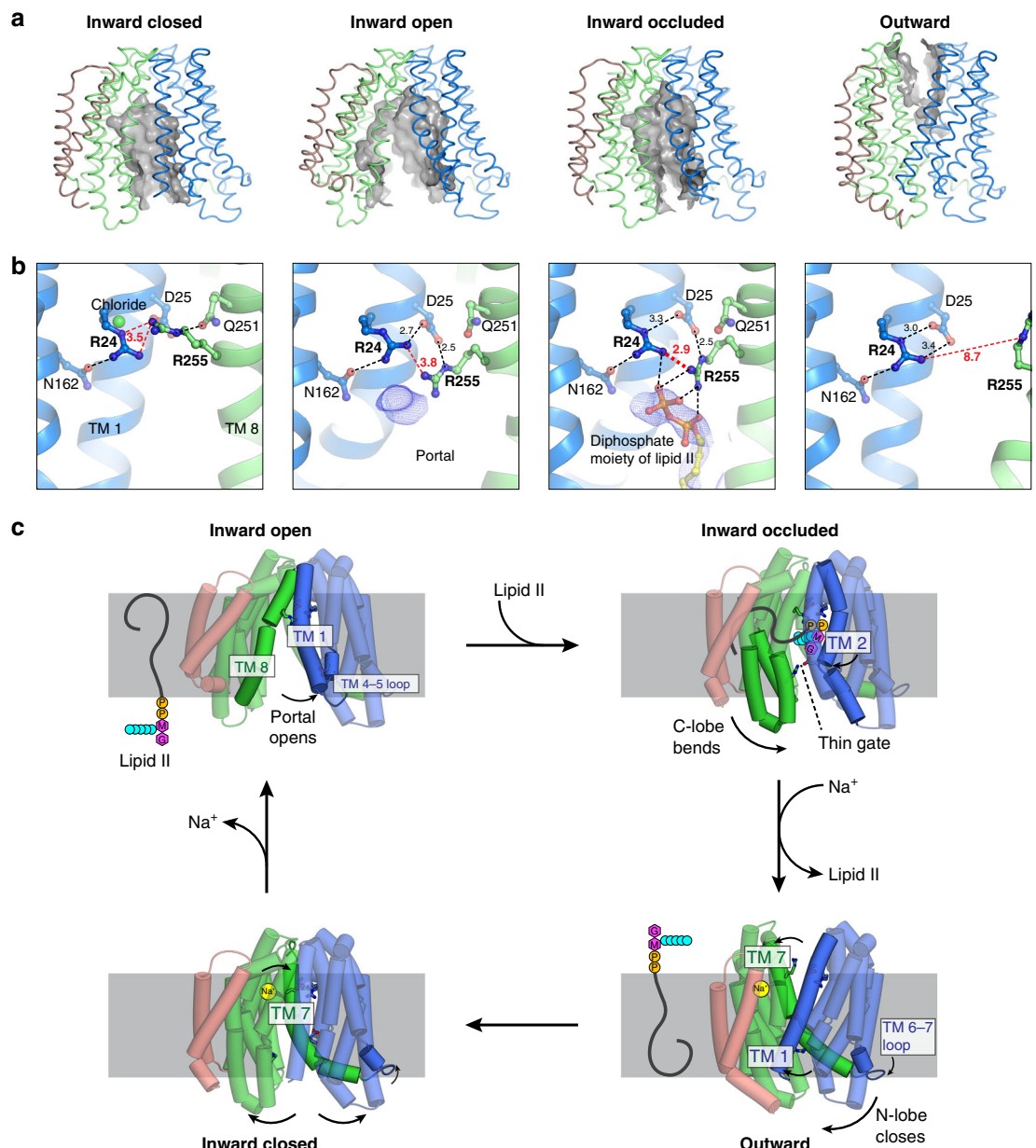

**Fig. 6** State-dependent conformational changes in the central cavity during the transport cycle. **a** The inward-facing cavity opens up in the inward-open conformation, and then extends deepest into the membrane in the inward-occluded conformation. In contrast, the outward-facing cavity is narrow and shallow. **b** Systematic rearrangement of the Arg24-Asp25-Arg255 triad in the cavity during the transport cycle. In the inward-occluded state, Arg24 and Arg255 are brought into unusual proximity (just 2.9 Å apart at the closest), with the unfavorable repulsion stabilized by electrostatic interactions with Asp25 and the diphosphate moiety of Lipid II. The lipid-diphosphate model was docked into final electron density (blue mesh, contoured to 1.0 σ) and was never used for refinement or map calculation. **c** Proposed model of the MurJ transport mechanism

## Methods

**Protein expression and purification for structural studies.** MurJ$_{TA}$ was expressed as described previously in 4 L culture of *E. coli* C41(DE3) (Lucigen) cells transformed with a modified pET26 plasmid encoding the construct His$_{10}$-MBP-PPX-MurJ-PPX-His$_{10}$[25]. Chloride was removed from all buffers used for cell lysis and protein purification, as detailed below. Cell pellets from each liter of culture were resuspended in 10 mL lysis buffer (50 mM HEPES-NaOH pH 8.0, 150 mM Na acetate) containing the following additives: 10 mM β-mercaptoethanol, 1 μg/mL leupeptin, 1 μg/mL pepstatin, 5 mU/mL aprotinin, 20 μg/mL deoxyribonuclease I, and 3 mM phenylmethylsulfonyl fluoride. Cells were disrupted by a Microfluidizer, and membrane proteins were extracted by stirring with 30 mM n-dodecyl-β-D-maltopyranoside (DDM, Anatrace) for 1 h at 4 °C. Upon centrifugation at 22,000 *g* for 30 min, the supernatant was rotated with 1.5 mL of TALON Cobalt resin (Clontech) for 30 min at 4 °C in the presence of 5 mM imidazole. Resin was recovered by centrifugation at 800 × *g* for 5 min and resuspended in 7.5 mL of wash buffer (50 mM HEPES-NaOH pH 8.0, 150 mM Na acetate, 15 mM imidazole,

1 mM DDM). The slurry was poured into a gravity column, and subsequently washed twice with 7.5 mL of the same buffer at 4 °C. Protein was eluted in 6 mL of wash buffer with imidazole added to 200 mM and incubated overnight at 4 °C with 40 μg/mL PreScission protease and 5 mM β-mercaptoethanol. PreScission protease-treated MurJ was exchanged to gel filtration buffer, containing 20 mM HEPES-NaOH pH 8.0, 150 mM Na acetate, 2 mM dithiothreitol, and 0.3 mM decyl maltose neopentyl glycol (DMNG, Anatrace), by 15-fold dilution and rotation for 15 min at 4 °C, and subsequently purified by gel filtration on a Superdex 200 10/300 column.

**Crystallization.** MurJ$_{TA}$ was crystallized by the lipidic cubic phase (LCP) method[45] in Lipid II-doped mesophase as described previously but with modifications[25]. Purified, chloride-free MurJ$_{TA}$ was concentrated to 20–25 mg/mL with a 50 kD molecular-weight-cutoff centrifugal filter, centrifuged at 16,000×*g* for 30 min at 4 °C, before mixing with molten Lipid II-doped monoolein (prepared as described previously[25]) in the standard 2:3 (w/w) water:lipid ratio using a twin-syringe setup.

MurJ-reconstituted LCPs were dispensed on 96-well glass sandwich plates as 130 nL drops by a Gryphon LCP robot (Art Robbins Instruments) and overlaid with 1 μL of precipitant solution from crystallization screens made in-house and/or the commercial MemMeso HT-96 screen (Molecular Dimensions).

The salt composition of the precipitant solution was the key determinant of which crystal form was obtained, which was also influenced by pH to a lesser extent. We did not notice any prep-to-prep variation in which crystal forms were obtained from each precipitant solution. Crystals of the inward-closed structure were obtained in 1 M NaCl, 50 mM Na acetate-HCl pH 4.6, 40% PEG400. Crystals of the inward-open structure were obtained in 200 mM ammonium sulfate, 50 mM Tris-HCl pH 8.5, 20% PEG400; or in 200 mM ammonium sulfate, 100 mM Na citrate tribasic-HCl pH 5.5, 40% PEG400. Crystals of the inward-occluded structure were obtained in 100 mM Na citrate tribasic-HCl pH 5.0, 40% PEG200 (MemMeso A4). Crystals of the outward structure were obtained in 1 M NaCl, 50 mM HEPES-NaOH pH 7.5, 20% PEG400; or in 100 mM MgCl₂, 100 mM NaCl, 100 mM HEPES-NaOH pH 7.0, 30% PEG500 DME (MemMeso E9); or in 100 mM MgCl₂, 100 mM NaCl, 100 mM MES-NaOH pH 6.0, 30% PEG500 MME (MemMeso F9). Crystals grew to full size in 4 weeks and were flash-frozen in liquid nitrogen without additional cryo-protectant.

**Data collection and structure determination.** We collected X-ray diffraction data at the NECAT 24-ID-C and 24-ID-E beamlines (Advanced Photon Source, Argonne National Laboratory) with a wavelength of ~0.98 Å. For the inward-occluded structure, we attempted to collect redundant data at a longer wavelength (1.65 Å) to locate anomalous difference Fourier electron density peaks corresponding to the diphosphate moiety of Lipid II, but without success likely due to the weak anomalous scattering power of phosphate and the inherent flexibility of Lipid II (~68 rotatable torsions). Nevertheless, the long-wavelength datasets were merged with the datasets collected at 0.98 Å for refinement as they were isomorphous and there was no noticeable radiation damage. Data were processed by XDS[46], and XDS-processed data from multiple isomorphous crystals were merged by POINTLESS and AIMLESS in BLEND[47–49]. Due to diffraction anisotropy in the high resolution shells, merged data were subjected to anisotropy correction in STARANISO[50]. Anisotropy corrected data were used for molecular replacement using our previously published structure of MurJ$_{TA}$ (PDB: 5T77) as the search model in PHASER[51]. For the outward structure, the search model was split into two fragments (residues 4–228 and residues 229–470) and a two-component search yielded a strong MR solution with log-likelihood gain of 1588 and translation function Z-score of 40. MR solutions were subjected to iterative rounds of model building in COOT[52] and refinement in PHENIX.refine[53]. Structures were refined to a final $R_{work}/R_{free}$ of 25.5/28.0% (inward-closed structure, 3.2 Å resolution), 25.4/27.8% (inward-open structure, 3.0 Å resolution), 22.9/25.8% (inward-occluded structure, 2.6 Å resolution), and 17.9/19.9% (outward structure, 1.8 Å resolution). The final models were refined to excellent geometry (>97% Ramachandran favored, 0% Ramachandran outliers, <1% rotamer outliers) and minimal clashes (clash-score < 2.5). Crystals contained either a single MurJ molecule in the asymmetric unit (inward-occluded and outward structures), or two MurJ molecules (inward-closed and inward-open structures).

Omit maps for Na⁺ were calculated in PHENIX.maps to the maximum resolution of the respective datasets. For the outward-facing structure, the Fo − Fc omit peak was 19 σ for the merged data used for structure refinement, or 17 σ for the single dataset crystallized in 1 M NaCl without Mg²⁺, with both maps calculated to 1.8 Å resolution. The Fo − Fc omit peak was 9 σ for the inward-occluded structure, and 7 σ for the inward-closed structure. Na⁺ coordination geometry was validated by the CheckMyMetal server[38]. Structural alignments and molecular graphics were created in PyMOL[54].

**Docking of Lipid II and molecular dynamics simulation.** The L-lysine form of Lipid II (1875 Da) was docked into the crystal structure of the inward-occluded conformation as described previously[25]. Briefly, Lipid II coordinates (including all hydrogens, protonation state set to neutral pH, i.e., protonated L-lysine but deprotonated phosphates/carboxylates) were generated from 2D geometry in PHENIX.eLBOW. Stereochemistry was rigorously checked and manually corrected in PHENIX.REEL, including configuration of all 16 chiral centers, planarity and cis/trans configuration of peptide and polyprenyl C=C double bonds, sugar pucker and glycosidic linkage. The remaining 68 torsions were set to be freely rotatable. The unresolved sidechains of Arg18 and Lys53 were built manually with the least-clashing ideal rotamer and subsequently treated as flexible for docking. Protein and ligand were prepared for docking in Autodock Tools, merging charges and removing non-polar hydrogens[55]. Docking was performed by Autodock Vina with a search space of 24 × 24 × 38 Å encompassing the central cavity, portal, and groove[56].

Molecular dynamics simulations starting from the docked model were set up as follows. The protein molecule from docking was processed by PDB2PQR to add all hydrogens (not just polar hydrogens, protonation state set to neutral pH) and all missing sidechains (not just those in the cavity)[57,58]. The protein molecule was embedded into a 1-Palmitoyl-2-oleoyl-sn-glycero-3-phosphoethanolamine (POPE) membrane with xy-dimensions of 140 × 120 Å with the membrane plugin (www.ks.uiuc.edu/Research/vmd/plugins/membrane) in VMD, which also removes any POPE molecules that overlap with the embedded protein[59]. This was performed

using the CHARMM all-hydrogen forcefield topology file for protein and lipids (top_all27_prot_lipid.inp)[60–62].

The Lipid II molecule in the docked conformation was processed by PHENIX.ReadySet (using the original restraints cif file from PHENIX.REEL) to add back all the hydrogens with protonation state set to neutral pH, and the Lipid II CHARMM forcefield topology file was generated by the SWISS-PARAM server (www.swissparam.ch)[63]. Lipid II was then embedded into the membrane by superposing the docked protein-Lipid II complex on to the embedded protein molecule in the membrane. The embedded protein molecule (with Arg18 and Lys53 in the docked conformation), Lipid II molecule (docked conformation), and POPE lipid bilayer were combined into a single file and solvated with TIP3P waters on both sides of the bilayer to an overall z-dimension of 100 Å using the Solvate plugin in VMD[59]. Total charge of the system was neutralized with addition of 150 mM NaCl by the Autoionize plugin in VMD[59]. Waters that are in the hydrophobic membrane interior (excluding those in the solvent-accessible cavity of MurJ) were manually deleted in PyMOL. There were no ions in the hydrophobic membrane interior. This solvated and ionized system (~140,600 atoms) was then processed by the AutoPSF plugin in VMD to generate the combined psf and pdb files for MD simulation, using the abovementioned CHARMM forcefield topology files (top_all27_prot_lipid.inp for protein/POPE and the SWISS-PARAM output for Lipid II)[59].

Molecular dynamics simulations were performed by NAMD[64], with reference to a tutorial manual (www.ks.uiuc.edu/Training/Tutorials/science/membrane/mem-tutorial.pdf). For all simulation runs, an integration time step of 1 fs was used and all X-H bonds were set to rigid (both bond lengths and angles). A periodic cell of 140 × 120 × 100 Å was used with Particle Mesh Ewald electrostatics evaluation[65], wrapping both waters and POPE lipid molecules that diffuse out of the boundaries back into the opposite side of the cell. To mimic the disorder in an actual membrane, the lipid tails of POPE molecules were first melted while everything else (including POPE headgroups) were fixed in position. Subsequently, gaps between the protein and POPE molecules were filled by a constrained equilibration run at 310 K and 1 atm, with energy minimization for 1000 steps and simulation for 100 ps. The follower parameters were also used: flexible cell, constant ratio, fixed atoms released, harmonic constraints on protein, keep_waters_out.tcl script to keep waters out from the bilayer). Finally, a production run was performed at 310 K and 1 atm for 100 ps releasing the constraints, and coordinates were recorded at 1 ps intervals. At the end of simulation, the stereochemistry of Lipid II was checked and no distortion was found.

**Expression and delipidation of cysteine mutants.** Because MurJ$_{TA}$ has no endogenous cysteine residues, single cysteine mutants could be engineered at either the cytoplasmic and/or periplasmic gates to probe the protein conformation in vitro in the absence or presence of sodium (Supplementary Table 4). One liter of each cysteine mutant was expressed as described for the structural studies. Sodium was removed from all cell lysis and protein purification buffers, replaced by either N-methyl-D-glucamine (NMDG) or potassium. Chloride was replaced by acetate in all buffers. The protein purification procedure in NMDG is detailed below. Cell pellets from each liter of culture were resuspended in 10 mL lysis buffer, which contained 50 mM Tris-acetate and 150 mM NMDG, pH 8. This buffer was made by dissolving the free base of both Tris and NMDG together, then adjusting to pH 8 with glacial acetic acid. The same additives were used in the lysis buffer as for structural studies, with the sole exception of β-mercaptoethanol being replaced by 1 mM Tris(2-carboxyethyl)phosphine (TCEP, Thermo Fisher). Cell disruption, membrane extraction, and cell debris removal were performed as described for structural studies. The supernatant was rotated with 0.5 mL of TALON Cobalt resin (Clontech) for 30 min at 4 °C in the presence of 5 mM imidazole. Resin was recovered by centrifugation at 800×g for 5 min and resuspended in 5 mL of wash buffer (50 mM Tris-acetate and 150 mM NMDG, pH 8, 15 mM imidazole, 1 mM TCEP, 1 mM DDM). The slurry was poured into a gravity column, and subsequently washed twice with 5 mL of the same buffer at 4 °C. Protein was eluted in 4 mL of wash buffer with imidazole added to 200 mM, then incubated overnight at 4 °C with 20 μg/mL PreScission protease without β-mercaptoethanol.

As the presence of any cardiolipin that is carried over from E. coli lysate could be inhibitory to MurJ function[20], we took measures to delipidate our MurJ samples. We exploited the high isoelectric point of MurJ$_{TA}$ (predicted to be 8.5–9.5) to delipidate MurJ$_{TA}$ by cation exchange. PreScission protease-treated proteins were diluted fivefold in buffer A (20 mM Tris-acetate pH 8.0, 1 mM DDM), treated with 1 mM TCEP, and passed through 0.5 mL of pre-equilibrated sulfopropyl(SP)-sepharose fast-flow resin (GE healthcare) in a gravity column twice at room temperature. The resin was washed with 5 mL each of 50 mM NMDG (16.7% buffer B), 100 mM NMDG (33.3% buffer B), and 150 mM NMDG (50% buffer B), before elution in 2 mL of 300 mM NMDG (100% buffer B, divided into four 0.5 mL fractions), all performed at room temperature. Buffer B contained 50 mM Tris-acetate pH 8.0, 300 mM NMDG, 1 mM DDM). Eluted protein was extremely pure (single band on SDS-PAGE) and mass spectrometry analysis indicated that all traces of cardiolipin had been successfully removed by the delipidation treatment. We did not perform delipidation for structural studies because the high purity came at the cost of drastically reduced recovery (10–30 μg/L culture). Eluted protein was concentrated to 3–5 μM with a 50 kD MW cutoff centrifugal filter before the assay.

**In vitro cysteine accessibility assay**. To probe the conformation of MurJ$_{TA}$ in vitro, single-cysteine mutants (no endogenous cysteines) were purified and delipidated in potassium acetate buffer as detailed above. Mutants were diluted to ~1 µM in reaction buffer containing 20 mM HEPES-KOH pH 7.5, 1 mM DDM, and 150 mM potassium acetate (buffer was added from 5× stock). Samples (9 µL) were equilibrated at either 20 or 60 °C for 10 min, then treated with 1 µL of methoxypolyethylene glycol 5000 maleimide (PEG-Mal, Sigma-Aldrich, 10× stock in water) to a working concentration of 0.5 mM. An equivalent volume of water was added to the negative control samples. Reactions were allowed to proceed at either 20 or 60 °C for 10 min, and subsequently quenched with 5 mM N-ethyl-maleimide (NEM, Sigma-Aldrich, 10× stock dissolved in ethanol) for 10 min at room temperature. Samples were mixed with 2× denaturing buffer (60 mM Tris-HCl pH 6.8, 8 M urea, 4% (w/v) SDS, 20% (v/v) glycerol, and 0.01% (w/v) bromophenol blue) in a 1:1 volume ratio, resolved by SDS-PAGE on a 10% polyacrylamide gel, and stained by Coomassie brilliant blue R-250.

**In vivo complementation assay**. MurJ$_{TA}$ mutants was generated by site-directed mutagenesis using standard recombinant DNA methods from MurJ$_{TA}$ wild-type in pEXT21 vector (Supplementary Table 4)[25]. The plasmid carrying MurJ$_{TA}$ (wild-type or mutants) were transformed into MurJ-depletion E.coli strain NR1154[8]. Growth of NR1154 is dependent on L-arabinose, while treatment with the anti-inducer D-fucose depletes endogenous MurJ. Transformants were selected with 80 µg/mL spectinomycin on LB-agar plates supplemented with 0.2% of arabinose. Overnight cultures diluted 1:1000 into fresh LB media with 80 µg/mL spectino-mycin and 0.2% of arabinose were grown to OD$_{600}$ of 0.6. Cultures were then split into two halves and a final concentration of 0.1 mM IPTG was added to one half. Cultures were further incubated for 3 h at 37 °C. Cells were collected by cen-trifugation and washed with fresh LB three times and normalized to an OD$_{600}$ of 0.2. Each culture was then diluted to 0.02, subjected to ten-fold serial dilution, and 5 µL were spotted onto LB-agar plates supplemented with either 0.05% fucose or 0.05% of fucose with 0.1 mM IPTG. To determine whether mutants were expres-sed, total membrane fractions were isolated and analyzed by western blot as was described previously[25]; primary antibody: mouse anti-FLAG (Sigma F3165); secondary antibody: horseradish peroxidase-conjugated goat anti-mouse (Invitro-gen #31430).

**In vivo Lipid II accumulation assay**. The plasmid carrying MurJ$_{TA}$ (wild-type, mutant, or empty vector) were transformed into E. coli NR1154 and transformants were selected with 80 µg/mL spectinomycin on LB-agar plates supplemented with 0.2% of arabinose. Overnight cultures were normalized to an OD$_{600}$ of 1.5 and diluted 1:10,000 into fresh LB media supplemented with 80 µg/mL spectinomycin and 0.05% of fucose. OD$_{600}$ were measured every hour after 2 h of shaking at 37 °C. After 6 h, cells were normalized to the lowest OD$_{600}$ of all samples and collected by centrifugation for mass spectrometry analysis.

Lipid extraction was performed using a modified Bligh-Dyer method[66]. Specifically, each cell pellet was re-suspended in 0.8 mL of 1× PBS (phosphate-buffered saline), and transferred in a 17 mL glass tube with a Teflon-lined cap, which is then followed by the addition of 2 mL of methanol and 1 mL of chloroform to create a single-phase Bligh-Dyer solution (chloroform/methanol/PBS, 1:2:0.8, v/v/v). The solution was vigorously vortexed for 2 min, followed by sonication in a water bath at room temperature for 20 min. After centrifugation at 3000×g for 10 min, the supernatant was transferred to a fresh glass tube and acidified by adding 50 µL of concentrated HCl (37%). After mixing, the acidified solution was converted into a two-phase Bligh-Dyer system by adding 1 mL of PBS and 1 mL of chloroform. After centrifugation at 3000 × g for 10 min, the lower phase was collected and dried under nitrogen gas. For Lipid II and Lipid X analysis, the dried lipid extracts were dissolved in a 100 µL solution consisting of chloroform/methanol (2:1, v/v), of which 20 µL was injected for normal phase LC/MS analysis.

Normal phase LC was performed on an Agilent 1200 Quaternary LC system equipped with an Ascentis Silica HPLC column (5 µm, 25 cm × 2.1 mm, Sigma-Aldrich, St. Louis, MO). Mobile phase A consisted of chloroform/methanol/aqueous ammonium hydroxide (800:195:5, v/v/v); mobile phase B consisted of chloroform/methanol/water/aqueous ammonium hydroxide (600:340:50:5, v/v/v); mobile phase C consisted of chloroform/methanol/water/aqueous ammonium hydroxide (450:450:95:5, v/v/v). The elution program consisted of the following: 100% mobile phase A was held isocratically for 2 min and then linearly increased to 100% mobile phase B over 14 min and held at 100% B for 11 min. The LC gradient was then changed to 100% mobile phase C over 3 min and held at 100% C for 3 min, and finally returned to 100% A over 0.5 min and held at 100% A for 5 min. The LC eluent (with a total flow rate of 300 µL/min) was introduced into the ESI source of a high resolution TripleTOF5600 mass spectrometer (Sciex, Framingham, MA). Instrumental settings for negative ion ESI and MS/MS analysis of lipid species were as follows: IS = −4500 V; CUR = 20 psi; GSI = 20 psi; DP = −55 V; and FP = −150 V. The MS/MS analysis used nitrogen as the collision gas. Data analysis was performed using Analyst TF1.5 software (Sciex, Framingham, MA). The levels of Lipid II, Lipid X, Phosphatidylethanolamine were quantified by using the integrated peak areas of their extracted ion chromatograms ($m/z$ 710.4 for the $[M-H]^-$ ion of Lipid X and 958.4 for the $[M-2H]^{2-}$ ion of Lipid II).

**Reporting summary**. Further information on experimental design is available in the Nature Research Reporting Summary linked to this article.

## Data availability
Data supporting the findings of this manuscript are available from the corresponding author upon reasonable request. A reporting summary for this Article is available as a Supplementary Information file. The source data underlying Fig. 5d and Supplementary Figs. 3a, 5, 6, 11 are provided as a Source Data file. Atomic coordinates and structure factors for the reported crystal structures are deposited in the Protein Data Bank under accession codes 6NC6, 6NC7, 6NC8, and 6NC9.

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

## Acknowledgements

We thank Natividad Ruiz (The Ohio State University) for sharing *E. coli* strains NR1154 and NR1157. We thank Ellene Mashalidis for critical manuscript reading. This work was supported by the National Institutes of Health (R01GM120594 to S.-Y.L.). Data for this study were collected at NE-CAT beamlines 24-ID-C and 24-ID-E at the Advanced Photon Source, which are funded by GM124165, RR029205, and OD021527.

## Author contributions

A.C.Y.K. solved the structures of MurJ$_{TA}$, performed in vitro accessibility assay, and conducted western blot analyses under the guidance of S.-Y.L. A.H. carried out the functional complementation assays and sample preparation for lipid mass spectrometry and western blot experiments under the guidance of S.-Y.L. Z.G. performed lipid mass spectrometry experiments. A.C.Y.K. and S.-Y.L. wrote the paper. All authors discussed the results and commented on the manuscript.

## Additional information

**Competing interests:** The authors declare no competing interests.

