## [Peer Review File · Nature Communications]

Reviewers' Comments:

Reviewer #1:

Remarks to the Author:

The authors have significantly improved their manuscript in response to previous reviews. The structures are excellent and a very important contribution. The proposed model makes sense, although the role of calcium is not clear, but the authors have toned down their statements sufficiently. However, there are still some issues for the authors to consider:

1) The conclusion that both MurJTA and MurJec use an alternating-access mechanism is not in dispute. There is evidence provided in this study and those of others supporting this conclusion. However, the conservation of some mechanistic details and roles assigned to various residues that the authors report here is in doubt when one examines published data. This is not to say that the mechanistic details that the authors propose for MurJTA are not correct, but that some of them are not conserved in MurJec. A similar scenario occurs in the related MATE transporters. The general mechanism and structure of MATE proteins is conserved, but specific mechanistic details are not. In fact, in the case of MurJ, Butler et al 2014 showed that while the role of positive charges provided by Arg18, Arg24, and Arg270 in MurJec is conserved across distant MurJ proteins, other charge requirements within the cavity are not. Those results support the idea that distant MurJ proteins might differ in some mechanistic details, while maintaining some fundamental features. Indeed, based on their analyses, Butler et al proposed that Arg18, Arg24, and Arg270 likely bind lipid II, as Kuk et al are now proposing.

A comparison of functional studies reported so far could be used to test whether all or some of the mechanistic details presented by this study by Kuk et al. are conserved across MurJ orthologs. The authors presented this type of comparison in Table S2. I agree with the authors that the high-throughput analysis by Zheng et al. has limitations with respect to the level of detail it provides, but it is a source of very useful qualitative information, while those from Kuk et al and Butler et al. provide more detailed analyses. However, Table S2 misses information that is relevant when interpreting the specific role and structural features of residues. For example, the Butler et al 2013 and 2014 papers reported that Cys substitutions at positions D25, E57, Q244, N250, and K368 were fully functional using complementation analysis and checking for growth defects in low osmolarity medium. Additionally, Zheng et al. reported effect of fitness using MutSeq for some but not all substitutions at those positions. Table S2 omits the Butler et al information, as well as the fact that some substitutions at those positions showed no defects in the Zheng et al study. Table S2 only represents data from Zheng et al showing that some substitutions lead to defects. This table should be modified to reflect all the data so that it does not misinform readers. If all the available data is taken into account, based on the function the authors propose for specific residues and the properties those residues need to have in order to perform that function, does it make sense that some of the substitutions reported by Butler et al (2013/2014) and Zheng et al do not have effects? (For example, if a salt bridge between two residues is required for a particular function, then Cys or Ala substitutions should interfere with that function).

2) The authors describe Glu57 and Arg352 as forming the thin gate and coming into proximity of ~4Å of each other in the inward-occluded structure. Could they please clarify the nature of interactions mediated by Glu57 and Arg352 to form the thin gate in the inward-occluded structure? This is relevant when interpreting the effect of substitutions in MurJ function as described above. Later in the manuscript, when describing the outward-facing structure, the authors state (page 10) that "Arg352 forms the thin gate with Glu57 in the inward-occluded structure, where it interacts with Ser61, Ser62, and the backbone carbonyl of Gly58 in the G/A-E-G-A helix break, suggesting that Arg352 plays a key role in conformational transition from the inward-facing to the outward-facing states." Fig. 4d shows these interactions occur in the outward-facing conformation, but the sentence indicates they occur in the inward-occluded structure. Do they occur in both? Please clarify.

3) The authors changed Arg352 to Ala and Gln, and concluded that both changes "led to loss of MurJTA function without loss of expression" (page 8, bottom). However, the blot in Fig. S6 clearly shows a reduction in the levels of the mutant proteins. It appears this reduction could be significant based on the image and the lack of quantification. Either provide quantification of biological replicates or rephrase the description of the results they show in Fig. S6 to indicate the apparent decrease.

4) Discussion, page 14 line 4-5: "We believe that the transport mechanism of lipid II flipping by MurJTA is conserved with that of other orthologs including MurJEC for the following reasons. (1) MurJTA complements MurJEC in *E. coli*". This is not a valid argument for their claim. Amj and Wzk are proteins that also complement *E. coli* but very likely use different mechanisms to translocate lipid II given how different they are with respect to structure.

We thank the reviewer for their thoughtful feedback on our manuscript. We revised our manuscript according to the reviewer's suggestions.

Point-by-point response

Reviewer #1 (Remarks to the Author):

1) The conclusion that both MurJ_{TA} and MurJ_{EC} use an alternating-access mechanism is not in dispute. There is evidence provided in this study and those of others supporting this conclusion. However, the conservation of some mechanistic details and roles assigned to various residues that the authors report here is in doubt when one examines published data. This is not to say that the mechanistic details that the authors propose for MurJ_{TA} are not correct, but that some of them are not conserved in MurJ_{EC}. A similar scenario occurs in the related MATE transporters. The general mechanism and structure of MATE proteins is conserved, but specific mechanistic details are not. In fact, in the case of MurJ, Butler et al 2014 showed that while the role of positive charges provided by Arg18, Arg24, and Arg270 in MurJ_{EC} is conserved across distant MurJ proteins, other charge requirements within the cavity are not. Those results support the idea that distant MurJ proteins might differ in some mechanistic details, while maintaining some fundamental features. Indeed, based on their analyses, Butler et al proposed that Arg18, Arg24, and Arg270 likely bind lipid II, as Kuk et al are now proposing. A comparison of functional studies reported so far could be used to test whether all or some of the mechanistic details presented by this study by Kuk et al. are conserved across MurJ orthologs. The authors presented this type of comparison in Table S2. I agree with the authors that the high-throughput analysis by Zheng et al. has limitations with respect to the level of detail it provides, but it is a source of very useful qualitative information, while those from Kuk et al and Butler et al. provide more detailed analyses. However, Table S2 misses information that is relevant when interpreting the specific role and structural features of residues. For example, the Butler et al 2013 and 2014 papers reported that Cys substitutions at positions D25, E57, Q244, N250, and K368 were fully functional using complementation analysis and checking for growth defects in low osmolarity medium. Additionally, Zheng et al. reported effect of fitness using MutSeq for some but not all substitutions at those positions. Table S2 omits the Butler et al information, as well as the fact that some substitutions at those positions showed no defects in the Zheng et al study. Table S2 only represents data from Zheng et al showing that some substitutions lead to defects. This table should be modified to reflect all the data so that it does not misinform readers. If all the available data is taken into account, based on the function the authors propose for specific residues and the properties those residues need to have in order to perform that function, does it make sense that some of the substitutions reported by Butler et al (2013/2014) and Zheng et al do not have effects? (For example, if a salt bridge between two residues is required for a particular function, then Cys or Ala substitutions should interfere with that function).

**** We completely agreed that the general mechanism of MurJ is conserved, but the specific details could be different between orthologues. We have never claimed that the detailed transport mechanism is identical between MurJ_{TA} and MurJ_{EC}, but we further clarified this**

point in the discussion section of the revision. In the discussion, we wrote “However, we also acknowledged that the specific mechanistic details could vary in different MurJ orthologues, as is common in transporters^{30,42} (Supplementary Table 2 and Supplementary Fig. 12). We caution against excessive mechanistic interpretation of previous functional results done on MurJ_{EC} as well as our complementation results on our MurJ_{TA} on the basis of our MurJ_{TA} structures because multiple factors (different types of experiments, different orthologues, and the lack of *in vitro* experiments) could make mechanistic interpretations of these experimental results non-trivial.”

As for the supplementary table 2, we included all the mutagenesis experiments done on MurJ_{EC} and MurJ_{TA} per reviewer’s suggestion and include new supplementary figure 12 to show structural features of residues by mapping them on our structure.

2) The authors describe Glu57 and Arg352 as forming the thin gate and coming into proximity of ~4Å of each other in the inward-occluded structure. Could they please clarify the nature of interactions mediated by Glu57 and Arg352 to form the thin gate in the inward-occluded structure? This is relevant when interpreting the effect of substitutions in MurJ function as described above. Later in the manuscript, when describing the outward-facing structure, the authors state (page 10) that "Arg352 forms the thin gate with Glu57 in the inward-occluded structure, where it interacts with Ser61, Ser62, and the backbone carbonyl of Gly58 in the G/A-E-G-A helix break, suggesting that Arg352 plays a key role in conformational transition from the inward-facing to the outward-facing states." Fig. 4d shows these interactions occur in the outward-facing conformation, but the sentence indicates they occur in the inward-occluded structure. Do they occur in both? Please clarify.

**** Arg352 forms a salt bridge with Glu57 in the inward-occluded state, but then changes its interaction partner to Ser61 and Ser62 in the outward-facing conformation, and thus the nature of interaction is now H-bonding. We acknowledge that our previous sentence is a bit confusing, and changed to the following: “Notably, Arg352 forms the thin gate with Glu57 in the inward-occluded structure, but now interacts with Ser61, Ser62, and the backbone carbonyl of Gly58 in the G/A-E-G-A helix break, suggesting that Arg352 plays a key role in conformational transition from the inward-facing to the outward-facing states”.**

3) The authors changed Arg352 to Ala and Gln, and concluded that both changes "led to loss of MurJ_{TA} function without loss of expression" (page 8, bottom). However, the blot in Fig. S6 clearly shows a reduction in the levels of the mutant proteins. It appears this reduction could be significant based on the image and the lack of quantification. Either provide quantification of biological replicates or rephrase the description of the results they show in Fig. S6 to indicate the apparent decrease.

**** we changed the sentence as follows to indicate the apparent reduction in the expression. “To test the importance of this thin gate to MurJ function, we mutated Arg352 to either Ala or Gln, both of which resulted in loss of complementation, albeit with some reduction of expression (Supplementary Figs 5 and 6). Together with our previous mutagenesis study on Glu57²⁵, this indicates that Glu57 and Arg352 are important for MurJ_{TA} function**

and/or folding, and suggests that electrostatic interaction might be the major contributor to thin gate stability.”

4) Discussion, page 14 line 4-5: “We believe that the transport mechanism of lipid II flipping by MurJTA is conserved with that of other orthologs including MurJEC for the following reasons. (1) MurJTA complements MurJEC in E. coli”. This is not a valid argument for their claim. Amj and Wzk are proteins that also complement E. coli but very likely use different mechanisms to translocate lipid II given how different they are with respect to structure.

**** We removed the complement argument.**

Reviewers' Comments:

Reviewer #1:

Remarks to the Author:

The authors have appropriately addressed my concerns. Very nice work!